# Expression Profile, Regulatory Network, and Putative Role of microRNAs in the Developmental Process of Asian Honey Bee Larval Guts

**DOI:** 10.3390/insects14050469

**Published:** 2023-05-16

**Authors:** Xiaoxue Fan, Wende Zhang, Sijia Guo, Leran Zhu, Yiqiong Zhang, Haodong Zhao, Xuze Gao, Haibin Jiang, Tianze Zhang, Dafu Chen, Rui Guo, Qingsheng Niu

**Affiliations:** 1College of Animal Sciences (College of Bee Science), Fujian Agriculture and Forestry University, Fuzhou 350002, China; imfanxx@163.com (X.F.); wdebee@163.com (W.Z.);; 2Apiculture Science Institute of Jilin Province, Jilin 132000, China; 3Apitherapy Research Institute of Fujian Province, Fuzhou 350002, China

**Keywords:** honey bee, *Apis cerana cerana*, miRNA, regulatory network, larva, gut

## Abstract

**Simple Summary:**

Currently, little is known about the regulatory role of miRNAs in the development of the Asian honey bee (*Apis cerana*) gut. On the basis of our previously gained small RNA-seq data, the expression pattern, regulatory network, and putative role of miRNAs in the developmental process of *A. cerana* worker larval guts were for the first time analyzed. DEmiRNAs were further analyzed. Here, 330, 351, and 321 miRNAs were identified in the 4-, 5-, and 6-day-old larval guts, respectively. Sequences of six miRNAs were verified using Sanger sequencing. Additionally, 15 and 10 differentially expressed miRNAs (DEmiRNAs) were, respectively, detected in the “Ac4 vs. Ac5” and“Ac5 vs. Ac6” comparison groups, and the target genes of these DEmiRNAs were involved in a series of GO terms and KEGG pathways relevant to growth and development, such as Notch and Wnt. miR-6001-y was induced to be activated in the development of larval guts. Based on RT-qPCR, we validated the expression levels of five randomly chosen DEmiRNAs. These findings lay a foundation for clarifying the developmental mechanism of the *A. cerana* worker larval guts.

**Abstract:**

MiRNAs, as a kind of key regulators in gene expression, play vital roles in numerous life activities from cellular proliferation and differentiation to development and immunity. However, little is known about the regulatory manner of miRNAs in the development of Asian honey bee (*Apis cerana*) guts. Here, on basis of our previously gained high-quality transcriptome data, transcriptome-wide identification of miRNAs in the larval guts of *Apis cerana cerana* was conducted, followed by investigation of the miRNAs’ differential expression profile during the gut development. In addition to the regulatory network, the potential function of differentially expressed miRNAs (DEmiRNAs) was further analyzed. In total, 330, 351, and 321 miRNAs were identified in the 4-, 5-, and 6-day-old larval guts, respectively; among these, 257 miRNAs were shared, while 38, 51, and 36 ones were specifically expressed. Sequences of six miRNAs were confirmed by stem-loop RT-PCR and Sanger sequencing. Additionally, in the “Ac4 vs. Ac5” comparison group, there were seven up-regulated and eight down-regulated miRNAs; these DEmiRNAs could target 5041 mRNAs, involving a series of GO terms and KEGG pathways associated with growth and development, such as cellular process, cell part, Wnt, and Hippo. Comparatively, four up-regulated and six down-regulated miRNAs detected in the “Ac5 vs. Ac6” comparison group and the targets were associated with diverse development-related terms and pathways, including cell, organelle, Notch and Wnt. Intriguingly, it was noticed that miR-6001-y presented a continuous up-regulation trend across the developmental process of larval guts, implying that miR-6001-y may be a potential essential modulator in the development process of larval guts. Further investigation indicated that 43 targets in the “Ac4 vs. Ac5” comparison group and 31 targets in the “Ac5 vs. Ac6” comparison group were engaged in several crucial development-associated signaling pathways such as Wnt, Hippo, and Notch. Ultimately, the expression trends of five randomly selected DEmiRNAs were verified using RT-qPCR. These results demonstrated that dynamic expression and structural alteration of miRNAs were accompanied by the development of *A. c. cerana* larval guts, and DEmiRNAs were likely to participate in the modulation of growth as well as development of larval guts by affecting several critical pathways via regulation of the expression of target genes. Our data offer a basis for elucidating the developmental mechanism underlying Asian honey bee larval guts.

## 1. Introduction

The honey bee is a kind of representative eusocial insect that plays an essential role in global ecological environment through pollination for a substantial quantity of crops and flowers. Additionally, honey bees can be domesticated for production of abundant and excellent api-products such as royal jelly, honey, propolis, beeswax, and bee venom [1]. Moreover, the honey bee is used as a research model to investigate growth and development [2], neurobiology [3], social behavior [4], host–pathogen interaction [5,6], and epigenetic regulation [7]. *Apis cerana* is a predominant bee species distributed across eastern, southern, and southeastern Asia and widely used in the beekeeping industry, with tremendous ecological and economic value [8]. In comparison with *Apis mellifera*, *A. cerana* has a subseries of advantages such as higher resistance to low temperature, greater ability to collect sporadic nectar sources, longer honey collection period, and stronger resilience to the ectoparasitic mite [9,10]. *Apis cerana cerana* is an endemic pollinator and commercial farming bee species in China and some other countries in Asia [11].

MicroRNAs (miRNAs) are a kind of small, non-coding RNA (ncRNA) molecule with a distribution range of 18~25 nucleotides (nt) in length, which is evolutionarily conserved and exerts pivotal functions in regulating considerable biological processes at the post-transcriptional level, such as development, metamorphosis, metabolism, immunity, and reproduction [12,13,14]. Increasing studies are suggestive of the participation of miRNAs in modulating the growth and development of various insects [15]. For instance, Shen et al. [15] found that injection of miR-2766 and miR-14 agomirs suppressed the levels of HaTH mRNA and protein and further resulted in defective pupation of *Helicoverpa armigera*. Wang et al. [16] verified that miR-282-5p in silkworm (*Bombyx mori*) regulated the expression of chitinase 5 (*chi5*), which was essential for the normal molting process, and observed that injection of miR-282-5p agomir led to the death of 40% of silkworms due to molting failure. By performing dual-luciferase assay and functional investigation, Zhang et al. [17] detected that miR-309a interacted with the transcription factor pannier (GATA-binding factor A/pnr) and then activated vitellogenin 2 (*Vg2*) and vitellogenin receptor (*VgR*) genes, which further facilitated the ovarian development. In honey bees, miRNAs have been shown to be regulators in developmental process [18], oviposition regulation [19], caste determination [20], immune defense [21], as well as organ development [22]. Previously, by performing small RNA-seq (sRNA-seq) and bioinformatics, we discovered 560 miRNAs in the *Apis mellifera ligustica* larval guts [23] and deciphered the expression profile as well as regulatory network of miRNAs across the developmental process of larval guts [24]. Currently, limited studies have shown that *A. cerana* miRNAs are potentially involved in response to fungi or viruses [25,26]. For example, Deng et al. [26] identified a total of 23 differently expressed miRNAs (DEmiRNAs) in *A. cerana* larvae upon CSBV infection and inferred that ame-miR-3759 may play a key part in melanin synthesis during larval response. Recently, we identified 435 miRNAs in the larval guts of *A. c. cerana*, the nominate subspecies of *A. cerana*, based on sRNA-seq and bioinformatics and further confirmed the sequences of six miRNAs and the expression of five miRNAs using molecular approaches [27].

The insect gut is a vital tissue responsible for digestion of food, absorption of nutrient, and immune defense [28]. Earlier studies regarding bee guts were mainly concentrated on the intestinal microorganisms [29,30,31]. However, few advancements in the developmental mechanism of the bee guts have been gained until now. Recently, we resolved the expression patterns, regulatory networks, and potential functions of piRNAs, another kind of small ncRNAs, in the larval guts of both *A. mellifera* and *A. cerana* [32,33]. However, little is known about the miRNA-regulated developmental mechanism underlying the bee gut at present.

Here, on the basis of our formerly obtained high-quality sRNA-seq datasets, miRNAs in *A. c. cerana* 4-, 5-, and 6-day-old larval guts were characterized using bioinformatics followed by the molecular validation; additionally, the differential expression pattern of miRNAs and corresponding regulatory networks during the developmental process of larval guts were investigated, with a focus on DEmiRNA-involved networks associated with development-associated signaling pathways as well as cellular and humoral immune. Our data can not only enrich our understanding of the development of the *A. cerana* larval gut but also lay a foundation for illustration of the mechanism underlying the DEmiRNA-modulated larval gut development.

## 2. Materials and Methods

### 2.1. Source of sRNA-seq Datasets

In our previous work, *A. c. cerana* 4-, 5-, and 6-day-old larvae were reared under laboratory conditions, and the larval gut tissues (Ac4 group: Ac4-1, Ac4-2, Ac4-3; Ac5 group: Ac5-1, Ac5-2, Ac5-3; Ac6 group: Ac6-1, Ac6-2, Ac6-3) were prepared following the established method [34], followed by cDNA library construction, deep sequencing utilizing sRNA-seq technology, and data quality control [27]. The high-quality raw data were deposited in the NCBI SRA database and linked to the BioProject number PRJNA395108.

### 2.2. Identification and Analysis of miRNAs

*A. c. cerana* miRNAs were identified according to our previously described protocol [27]: The clean reads were mapped to the reference genome of *A. cerana* (Assembly ACSNU-2.0), and the mapped clean reads were then aligned to GenBank (https://www.ncbi.nlm.nih.gov/genbank/, accessed on 2 March 2021) and Rfam (11.0) (http://rfam.xfam.org/, accessed on 2 March 2021) databases to remove small ncRNA including rRNA, scRNA, snoRNA, snRNA, and tRNA; further, clean tags of the reference genome were compared with known miRNA precursor sequences in the miRbase database (http://www.miRBase.org, accessed on 2 March 2021) to identify known miRNAs, while novel miRNAs were identified according to whether miRNAs had classical stem-loop structures. For each miRNA, the expression level was normalized to tags per million (TPM) based on the following formula: TPM = T × 10^6^/N (T refers to miRNA clean reads; N refers to total sRNA clean reads).

### 2.3. Validation of miRNAs by Stem-Loop RT-PCR and Sanger Sequencing

Six miRNAs (miR-14-y, bantam-y, miR-184-y, miR-9-z, miR-8-y, and miR-275-y) were randomly selected for RT-PCR. Specific stem-loop primers and forward primers (F) as well as universal reverse primers (R) for each of these DEmiRNAs were designed using Primer Premier 6 [35] (Appendix A) and then synthesized by Sangon Biotech Co., Ltd. (Shanghai, China). Using an RNA extraction kit (Promega, China), total RNA was extracted from 4-, 5-, and 6-day-old *A. c. ceranae* larvae guts (N = 3) and then divided into two portions. Next, reverse transcription was performed with stem-loop primers, and the resulting cDNA was used as template for PCR. On basis of the method described by Wu et al. [36], PCR amplification was conducted and subjected to 1.8% agarose gel electrophoresis, followed by extraction of the expected fragment, ligation to the pESI-T vector (Yeasen, China), and transformation into *Escherichia coli* DH5α-competent cells. After microbial PCR, 1 mL of bacterial solution with a positive signal was sent to Sangon Biotech Co., Ltd. (Shanghai, China) for single-end Sanger sequencing.

### 2.4. Investigation of DEmiRNAs

For each miRNA, the fold change in expression level between “Ac4 vs. Ac5” and “Ac5 vs. Ac6” comparison groups was determined according to the following formula: (TPM in Ac5)/(TPM in Ac4) or (TPM in Ac6)/(TPM in Ac5). Based on the standard of *p*-value ≤ 0.05 and |log_2_(Fold change)| ≥ 1, DEmiRNAs in “Ac4 vs. Ac5” and “Ac5 vs. Ac6” comparison groups were filtered out. Upregulated miRNAs refers to expression level of miRNAs between two groups with log_2_(Fold change) > 1 and downregulated miRNAs with log_2_(Fold change) < −1. By using the OmicShare platform (https://www.omicshare.com/, accessed on 5 March 2021), Venn analysis as well as expression clustering of the DEmiRNAs in these comparison groups were performed.

### 2.5. RNA-seq Data Source

Our group previously prepared the 4-, 5-, and 6-day-old larval guts of *A. c. cerana* followed by construction of strand-specific cDNA libraries and RNA-seq, and the produced raw reads were subjected to strict quality control to generate high-quality clean reads [34,37], which were used to conduct target prediction of DEmiRNAs in this current work. The raw data derived from RNA-seq are available in the NCBI SRA database under the following accession number: SRA456721.

### 2.6. Prediction and Annotation of DEmiRNA-Targeted mRNAs

Following the procedure outlined by Chen et al. [25], target mRNAs of DEmiRNAs were predicted by utilizing a collaboration of three software, including MiRanda (V3.3a) [38], RNAhybrid (V2.1.2) [39], and TargetFind [40], and the intersection was deemed as reliable targets. Next, these targets were mapped to the Gene Ontology (GO) (http://www.geneontology.org/, accessed on 5 March 2021) and Kyoto Encyclopedia of Genes and Genomes (KEGG) (https://www.kegg.jp, accessed on 5 March 2021) databases to gain corresponding function and pathway annotations by using Blast tool with default parameters, followed by drawing chord diagrams and bubble diagrams with related tools in the OmicShare platform.

### 2.7. Construction and Investigation of DEmiRNA-DEmRNA Regulatory Networks

Based on the predicted targeting relationships with a threshold of binding free energy less than −15 kcal/mol, regulatory networks between *A. c. cerana* DEmiRNAs in “Ac4 vs. Ac5” and “Ac5 vs. Ac6” comparison groups and target mRNAs were constructed and then visualized by Cytoscape v.3.2.1 software [41] with default parameters. In order to explore the miRNA-mediated process of growth and development as well as the immune process during the stage of 4-day-old to 6-day-old larvae, following the KEGG database annotations, targets relevant to development-associated signaling pathways as well as cellular and humoral immune pathways were filtered out, followed by construction and visualization of corresponding regulatory networks utilizing the method mentioned above.

### 2.8. RT-qPCR Verification of DEmiRNAs

The cDNA was synthesized as described in 2.3 for qPCR reaction, which was performed on QuanStudio 3 Fluorescence quantitative PCR instrument (ABI, Los Angeles, CA, USA). The reaction system (20 μL) contained 10 μL of SYBR green dye, 1 μL of forward and reverse primers (2.5 μmol/L), 1 μL of cDNA template, and 7 μL of DEPC water. The reaction conditions were set as follows: 95 °C pre-denaturation for 5 min; 95 °C denaturation for 30 s; 60 °C annealing and extension for 30 s; and a total of 40 cycles each group of qPCR reaction, and the experiment was set for repeating three times. The U6 gene (GenBank accession number: LOC108002035) was used as internal reference. The relative expression level of each DEmiRNA was calculated using the 2^−∆∆Ct^ method [42]. Each experiment was carried out with at least three samples in parallel and was repeated three times. The primers used in this work are detailed in the Appendix A.

### 2.9. Statistical Analysis

Fisher’s exact test was performed to filter out the significant KEGG pathways with the R package 3.3.1 (R Development Core Team, https://www.r-project.org/, accessed on 5 March 2021). The qPCR data are shown as mean ± standard deviation (SD) and subjected to Student’s *t*-test using SPSS 16.0 (IBM, Armonk, NY, USA) and GraphPad Prism v8.0 software (GraphPad, San Diego, CA, USA, https://www.graphpad.com/features, accessed on 5 May 2022). *p* < 0.05 was considered statistically significant.

## 3. Results

### 3.1. Analysis of miRNAs in the Larval Guts of A. c. cerana

Here, 330, 351, and 321 miRNAs were identified in the 4-, 5-, and 6-day-old larval guts, respectively. Venn analysis suggested that as many as 257 miRNAs were shared by the 4-, 5-, and 6-day-old larval guts, whereas 38, 51, and 36 ones were specific for each group (Figure 1A). In addition, expression clustering analysis showed that the identified *A. c. cerana* miRNAs presented various trends during the process of development of the larval guts (Figure 1B). Further investigation indicated that one DEmiRNA, namely miR-6001-y, displayed a continuous up-regulation trend in the larval guts, with TPM values of 508.86, 3961.65, and 9390.71, respectively. PCR amplification and Sanger sequencing of six randomly selected miRNAs demonstrated that their sequences were consistent with the predicted sequences based on sRNA-seq data, which verified the authenticity of the *A. c. cerana* miRNAs identified in this work (Figure 1C).

### 3.2. Structural Characteristics of miRNAs in 4-, 5-, and 6-Day-Old Larval Guts

The length distribution of miRNAs in Ac4, Ac5, and Ac6 groups ranged from 18 nt to 25 nt; the most enriched length was 22 nt, followed by 23 nt and 18 nt (Figure 2A). In addition, for miRNAs with length ranging from 18 to 25 nt, the first base is always biased toward U (Figure 2B), and miRNAs in the aforementioned three groups had similar bias at each base (Figure 2C).

### 3.3. Expression Pattern of miRNAs during the Development Process of the A. c. cerana Larval Guts

In the Ac4 vs. Ac5 comparison group, seven up-regulated and eight down-regulated miRNAs were detected; of these, the three most up-regulated miRNAs were miR-1672-x (log_2_FC = 8.41, *p* < 0.01), novel-m0032-5p (log_2_FC = 6.79, *p* < 0.01), and miR-6001-y (log_2_FC = 2.96, *p* < 0.05), while the three most down-regulated ones were novel-m0016-5p (log_2_FC = −9.25, *p* < 0.01), novel-m0016-5p (log_2_FC = −9.25, *p* < 0.01), and miR-6941-x (log_2_FC = −3.74, *p* < 0.05), respectively (Figure 3A). Comparatively, there were four up-regulated and six down-regulated miRNAs in the Ac5 vs. Ac6 comparison group (Figure 3B), and of these, the most up-regulated miRNA was miR-342-y (log_2_FC = 3.38, *p* < 0.05), followed by miR-6001-y (log_2_FC = 1.25, *p* < 0.05) and novel-m0005-3p (log_2_FC = 1.16, *p* < 0.05), whereas the most down-regulated miRNA was miR-26-x (log_2_FC = −9.41, *p* < 0.01), followed by miR-21-x (log_2_FC = −8.73, *p* < 0.01) and miR-1-z (log_2_FC = −2.32, *p* < 0.05) (Figure 3B).

### 3.4. Investigation and Annotation of DEmiRNA-Targeted mRNAs

DEmiRNAs in the Ac4 vs. Ac5 comparison group were predicted to target 5041 mRNAs, which were annotated to 18 biological-process-associated GO terms such as cellular process (any process that is carried out at the cellular level, but not necessarily restricted to a single), metabolic process (the chemical reactions and pathways, including anabolism and catabolism, by which living organisms transform chemical substances), and biological regulation (any process that modulates a measurable attribute of any biological process, quality or function); 15 cellular component-associated terms such as cell, cell part (refers to various cellular components), and membrane; and 11 molecular-function-associated terms such as binding (refers to binding relationship between different biological molecules), catalytic activity (catalysis of a biochemical reaction at physiological temperatures), and transporter activity (enables the directed movement of substances into, out of, or within a cell or between cells). In contrast, DEmiRNAs within the Ac5 vs. Ac6 comparison group could target 3727 mRNAs, which were annotated to 18 biological-process-related functional terms such as cellular process and metabolic process, 17 cellular-component-associated terms such as cell part and membrane, and 11 molecular-function-associated terms such as catalytic activity and transporter activity (Figure 4).

Additionally, the targets in the Ac4 vs. Ac5 comparison group were shown to be engaged in 130 KEGG pathways associated with metabolism, organismal systems, cellular processes, environmental information processing, and genetic information processing, including the biosynthesis of amino acids, dorso-ventral axis formation, endocytosis, phosphatidylinositol signaling system, and RNA degradation (Figure 5A). In the Ac5 vs. Ac6 comparison group, the targets were implicated in 107 pathways such as the biosynthesis of amino acids, dorso-ventral axis formation, endocytosis, Jak-STAT signaling pathway, and RNA degradation (Figure 5B).

### 3.5. DEmiRNA-mRNA Regulatory Network Engaged in the Development of A. c. cerana Larval Guts

In the Ac4 vs. Ac5 comparison group, novel-m0032-5p had the highest number of targets (2743), followed by miR-6001-y (2042 targets) and novel-m0049-3p (630 targets); among these targets, each of the 12 mRNA (XM_017052480.1, XM_017052481.1, XM_017052482.1, XM_017052624.1, XM_017052627.1, XM_017052629.1, XM_017054924.1, XM_017058051.1, XM_017062539.1, XM_017062547.1, XM_017062549.1, and XM_017062558.1) was targeted by the five DEmiRNAs. Additionally, miR-6001-y in the Ac5 vs. Ac6 comparison group potentially targeted the most mRNAs (2042), followed by miR-26-x (599 mRNAs) and miR-1-z (466 mRNAs); among these targets, each of the seven mRNA (XM_017048298.1, XM_017048300.1, XM_017057380.1, XM_017057382.1, XM_017064695.1, XM_017064696.1, and XM_017064697.1) was detected to be targeted by six DEmiRNAs.

Further analysis demonstrated that nine DEmiRNAs in the Ac4 vs. Ac5 comparison group could target 43 mRNAs annotated in development-associated signaling pathways including Wnt, Hippo, and Notch (Figure 6A). In contrast, eight DEmiRNAs in the Ac5 vs. Ac6 comparison group could target 31 mRNAs involved in the Wnt, Hippo, and Notch signaling pathways (Figure 6B).

### 3.6. DEmiRNA-mRNA Regulatory Network Involved in the Cellular and Humoral Immune of A. c. cerana Larval Guts

In the Ac4 vs. Ac5 comparison group, six DEmiRNAs could target four mRNAs annotated in the humoral immune-related signaling pathway Jak-STAT, and nine DEmiRNAs could target 61 mRNAs annotated in cellular immune-related signaling pathways including endocytosis, phagosome, and lysosome (Figure 7A). In the Ac5 vs. Ac6 comparison group, four DEmiRNAs could target seven mRNAs annotated in Jak-STAT, while seven DEmiRNAs could target 34 mRNAs annotated in endocytosis, phagosome, and lysosome (Figure 7B).

### 3.7. RT-qPCR Detection of DEmiRNAs

RT-qPCR detection showed that the expression trends of five randomly selected DEmiRNAs were in accordance with those in the sequencing results (Figure 8), confirming the trustworthiness of the transcriptome datasets employed in the present study.

## 4. Discussion

The larval stage of a honey bee lasts until it is six days old. It is hard to dissect intact gut tissues of 1- and 2-day-old bee larvae, which are very small and vulnerable. Hence, 3-day-old larvae were previously removed from the combs and placed in the 24-well culture plates to adapt to the lab conditions for 24 h, and 4-, 5-, and 6-day-old larval guts were selected for sequencing and investigation [27]. Moreover, we previously conducted strand-specific cDNA library-based RNA-seq of 4-, 5-, and 6-day-old larval guts, and the generated datasets included mRNAs and ncRNAs [34,37]. To maintain uniform larval ages in different sequencing projects enables researchers to analyze the interaction relationship between mRNAs and various ncRNAs such as miRNAs, piRNAs, lncRNAs, and circRNAs.

Here, as many as 257 miRNAs were shared in the *A. c. cerana* Ac4, Ac5, and Ac6 groups, indicating that the majority of miRNAs were co-expressed in 4-, 5-, and 6-day-old larval guts. It is inferred that these shared miRNAs play indispensable roles in the development of larval guts. Additionally, there were 38, 51, and 36 miRNAs specifically expressed in guts of 4-, 5-, and 6-day-old larvae, which is suggestive of their putative functions in the larval guts at different days of the larval stage. Previously, we identified 560 miRNAs in the larval guts of *Apis mellifera ligustica*, a subspecies of *A. mellifera*, among which 331 (59%) miRNAs were shared by the 4-, 5-, and 6-day-old larval guts, whereas the quantities of specific miRNAs were 44, 38, and 88, respectively [23]. Together, these results indicated that expression of the majority of miRNAs was essential for the development of the larval guts of both *A. c. cerana* and *A. m. ligustica*.

In this current work, following investigation of the structural characteristics, we observed that miRNAs in the 4-, 5-, and 6-day-old larval guts ranged from 18 nt to 25 nt in length, and the first base is always biased toward “U”, which is consistent with miRNAs identified in *Equus caballus* [43], *Bombyx mori* [44], and *Crassostrea gigas* [45]. This was indicative of the high conservation of structural characteristics of miRNAs in various animals and plants. A putative explanation for the “U” bias of first base was that it is more helpful in forming a protein complex with AGO protein, binding to the target gene, and regulating the gene expression [46].

MiRNAs were capable of regulating gene expression by targeting 3′ UTR or the coding sequence of corresponding mRNAs, further modulating numerous life activities [47,48]. In the present study, 15 and 10 DEmiRNAs were predicted to target 5041 and 3727 mRNAs in Ac4 vs. Ac5 and Ac5 vs. Ac6 comparison groups, respectively, involving 44 and 46 GO terms such as binding, cellular process, and catalytic activity as well as 130 and 107 KEGG pathways such as biosynthesis of amino acids, dorso-ventral axis formation, and endocytosis. The results demonstrated that corresponding DEmiRNAs may play regulatory roles in extensive aspects during the developmental process of the *A. c. cerana* larval guts.

Growth and development of organisms depend on the concerted action of multiple conserved signaling pathways [49,50]. Here, we observed that 10 DEmiRNAs (miR-6001-y, miR-3759-y, miR-3793-x, etc.) in the Ac4 vs. Ac5 comparison group and eight DEmiRNAs (miR-6001-y, miR-1-z, miR-8117-y, etc.) in the Ac5 vs. Ac6 comparison group could target 21 mRNAs and 16 mRNAs relevant to Wnt signaling pathway. In *Drosophila*, two branches of the Wnt signaling pathway are required for maintenance and regeneration of intestinal stem cells [51,52]. The DEmiRNA-mediated regulation of Wnt in *A. c. cerana* larvae gut development is expected to be further explored. The Hippo signaling pathway is conserved among various organisms and participates in the regulation of the size of organs [53]. In this current work, seven miRNAs (miR-6001-y, miR-3793-x, novel-m0016-5p, etc.) in the Ac4 vs. Ac5 comparison group and seven miRNAs (miR-6001-y, miR-1-z, miR-8117-y, etc.) in the Ac5 vs. Ac6 comparison group could target 20 and 16 mRNAs associated with the Hippo signaling pathway. The results indicated that corresponding DEmiRNAs may participate in the modulation of the development of larval guts via regulation of Hippo signaling pathways.

Gsk3β, an evolutionarily conserved serine/threonine kinase in all eukaryotes, regulates diverse biological processes from growth and metabolism to cell cycle and apoptosis [54,55]. The *Drosophila* Shaggy (Sgg) protein is the homologue of glycogen synthase kinase 3β (Gsk3β) [56]. Wu et al. [57] discovered that loss of *sgg* in *Drosophila* gave rise to Yorkie inhibition and downregulation of target genes in the Hippo signaling pathway, and *sgg* regulated the growth of eyes and wings via the Hippo signaling pathway. Here, kinase shaggy-like (XM_017057538.1) was found to be targeted by three DEmiRNAs (miR-6001-y, miR-1638-x, and novel-m0032-5p) in the Ac4 vs. Ac5 comparison group and two DEmiRNAs (miR-6001-y and miR-1-z) in the Ac5 vs. Ac6 comparison group. It demonstrated that *Sgg* may be continuously regulated by *A. c. cerana* larvae DEmiRNAs, and furthermore, miR-6001-y might participate in regulating *Sgg* to mediate growth.

The Son of sevenless (Sos) protein has been suggested to function as a guanine nucleotide transfer factor for Ras and to interact with the receptor tyrosine kinase Sevenless via the Drk protein, a homolog of mammalian Grb2 [58]. Previous studies showed that SOS1-knockout mutants were embryonic lethal [59]. Here, we noticed that the *sos* (XM_017058556.1) was a common target for miR-6001-x, miR-1638-x, and novel-m0049-3p in the Ac4 vs. Ac5 comparison group as well as miR-1-z in the Ac5 vs. Ac6 comparison group. In summary, these findings suggested that the aforementioned five DEmiRNAs may well regulate the growth and development of *A. c. cerana* larval guts via regulating *sos* genes’ expression. However, more efforts are needed to verify their interactions and underlying regulatory mechanisms.

In *Bombus terrestris*, the expression levels of both Bte-miR-6001-5p and Bte-miR-6001-3p were higher in queen- than in worker-destined late-instar larvae [60]. In *A. mellifera*, ame-miR-6001-5p has been suggested to play a regulatory part in ecdysone secretion and caste differentiation [61,62]. In this work, miR-6001-y was found to upregulate in both Ac4 vs. Ac5 and Ac5 vs. Ac6 comparison groups, which is suggestive of the activation of miR-6001-y during the development process of larval guts. In addition, we observed that miR-6001-y putatively targeted as many as 2042 mRNAs, involving Wnt, Hippo, and Jak-STAT signaling pathways, suggesting the involvement of miR-6001-y in adjusting these three signaling pathways of great importance. Additional work such as overexpression and knockdown of miR-6001-y in the larval guts via feeding method, following our previously established protocol [36], is required to decipher the function and action mechanism of miR-6001-y.

Different ncRNAs such as lncRNAs and circRNAs, which contain miRNA response elements, are able to interact with miRNAs via competing endogenous networks to further regulate expression of downstream genes [63]. In insects, ncRNA-involved ceRNA networks were shown to affect and development [16], metabolism [64], and immunity [65]. For example, Wang et al. [66] discovered that the nucleus-enriched lncR26319 in silkworm was able to regulate Endophilin A (*EndoA*), a member of the endophilin family of endocytic proteins, through competitive binding to miR-2834; the lncR26319-miR-2834-EndoA axis was required for Vg endocytosis. Our group previously discovered that 29 up-regulated lncRNAs and 14 up-regulated circRNAs in *A. m. ligustica* workers’ midguts potentially targeted ame-miR-6001-3p [67,68]. Findings from this work and our previous studies demonstrate that miR-6001 as a hub within ceRNA regulatory networks likely exerts a critical function in modulating the larval guts’ growth and development and workers’ midguts in both *A. c. cerana* and *A. m. ligustica*. Recently, we established the platforms for dissecting the functions of miRNA [36] and lncRNA [69] in bee larval guts. Hence, to identify candidate lncRNAs and circRNAs targeted by ace-miR-6001 and investigate the regulatory functions of the lncRNA-ace-miR-6001 axis and circRNA-ace-miR-6001 axis in the process of the development of larval guts is an interesting and promising direction in the near future.

## 5. Conclusions

In conclusion, the dynamic expression and structural alteration of miRNAs were accompanied by the development of *A. c. cerana* larval guts; DEmiRNAs were likely to participate in the modulation of the growth and development of larval guts by regulating the expression of target genes via affecting an array of crucial pathways including Wnt and Hippo; miR-6001-y is a potential key regulator in the development of larval guts by ceRNA mechanism or silencing genes. Our data offer a basis for elucidating the developmental mechanism underlying Asian honey bee larval guts.

## Figures and Tables

**Figure 1 insects-14-00469-f001:**
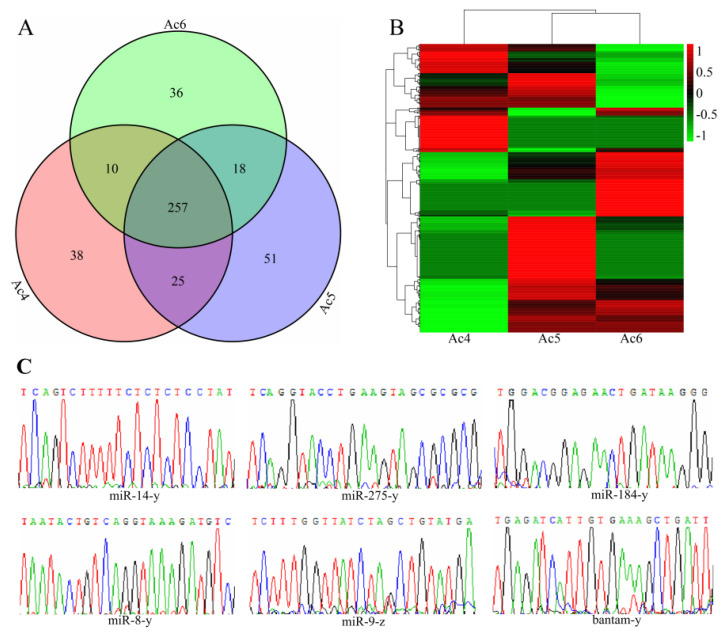
Expression analysis and molecular validation of miRNAs identified in the *A. c. cerana* 4-, 5-, and 6-day-old larval guts. (**A**) Venn analysis of miRNAs discovered in the 4-, 5-, and 6-day-old larval guts; (**B**) expression clustering of miRNAs expressed in the 4-, 5-, and 6-day-old larval guts; (**C**) Sanger sequencing of six miRNAs shared by the 4-, 5-, and 6-day-old larval guts.

**Figure 2 insects-14-00469-f002:**
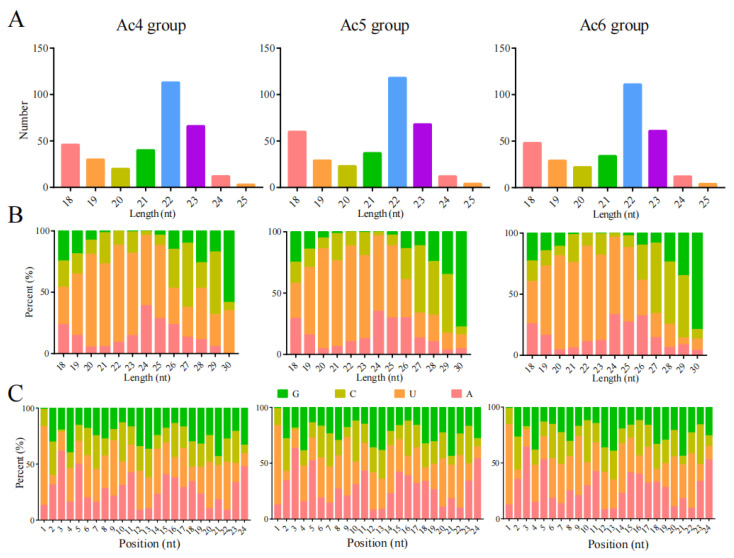
Structural characteristics of miRNAs in *A. c. cerana* 4-, 5-, and 6-day-old larval guts. (**A**) Length distribution of miRNAs; (**B**) first nucleotide bias of miRNAs; (**C**) bias of each nucleotide of miRNAs.

**Figure 3 insects-14-00469-f003:**
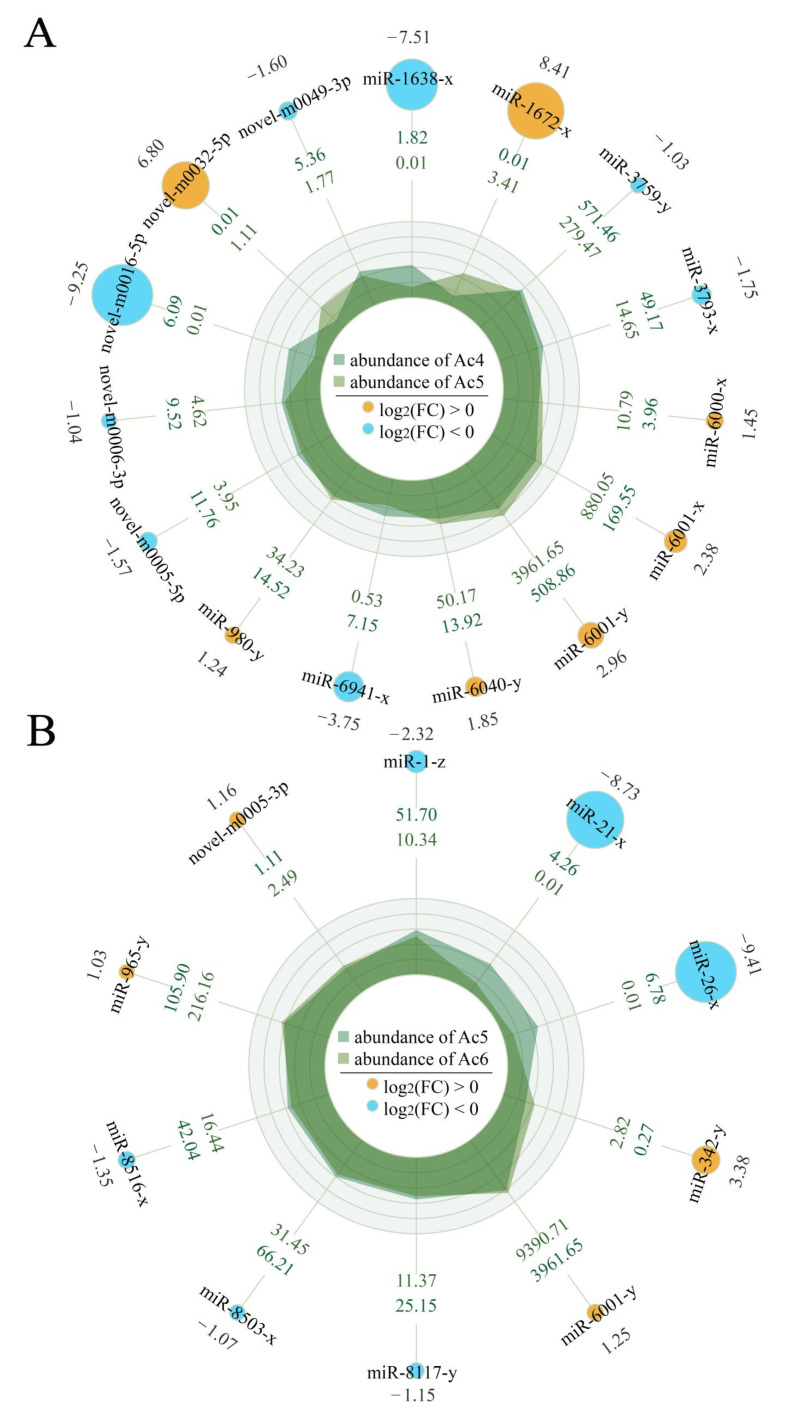
Radar maps of DEmiRNAs in the Ac4 vs. Ac5 (**A**) and Ac5 vs. Ac6 (**B**) comparison groups. Blue circles represent downregulation, while orange circles represent upregulation. The larger the circles, the greater the difference.

**Figure 4 insects-14-00469-f004:**
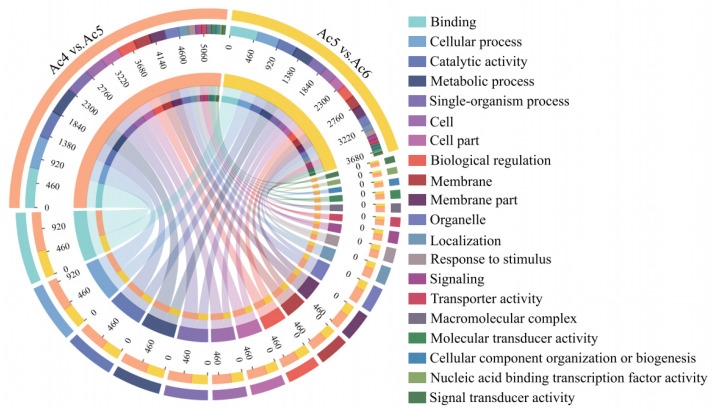
GO terms annotated by the DEmiRNA-targeted mRNAs in the Ac4 vs. Ac5 and Ac5 vs. Ac6 comparison groups. Different colors indicate different functional terms. The numbers inside the chord diagram indicate target mRNAs annotated to corresponding terms.

**Figure 5 insects-14-00469-f005:**
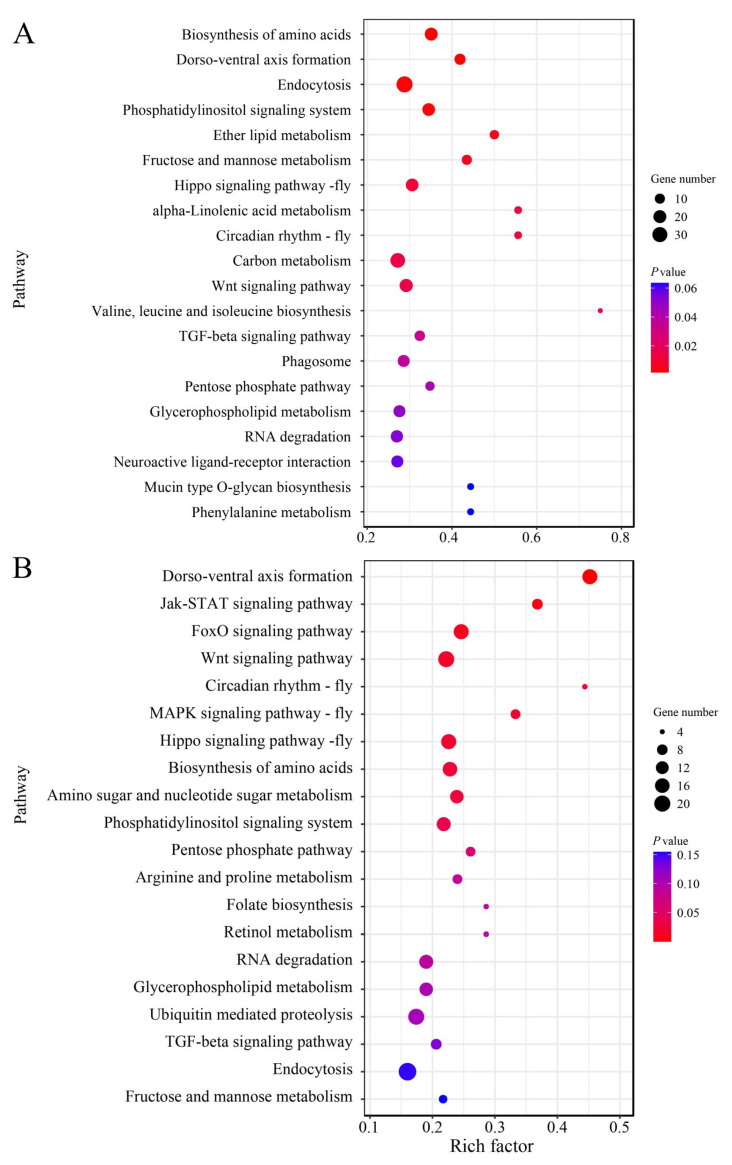
KEGG pathways annotated by the targets of DEmiRNAs in the Ac4 vs. Ac5 (**A**) and Ac5 vs. Ac6 (**B**) comparison groups. Bubbles indicate target mRNAs enriched in corresponding pathways; the larger the bubbles, the greater the amount of target mRNAs, and the smaller the bubbles, the fewer the target mRNAs. Different colors represent the *p*-values of different pathways; the red color indicates higher *p*-values, while the purple color indicates lower *p*-values.

**Figure 6 insects-14-00469-f006:**
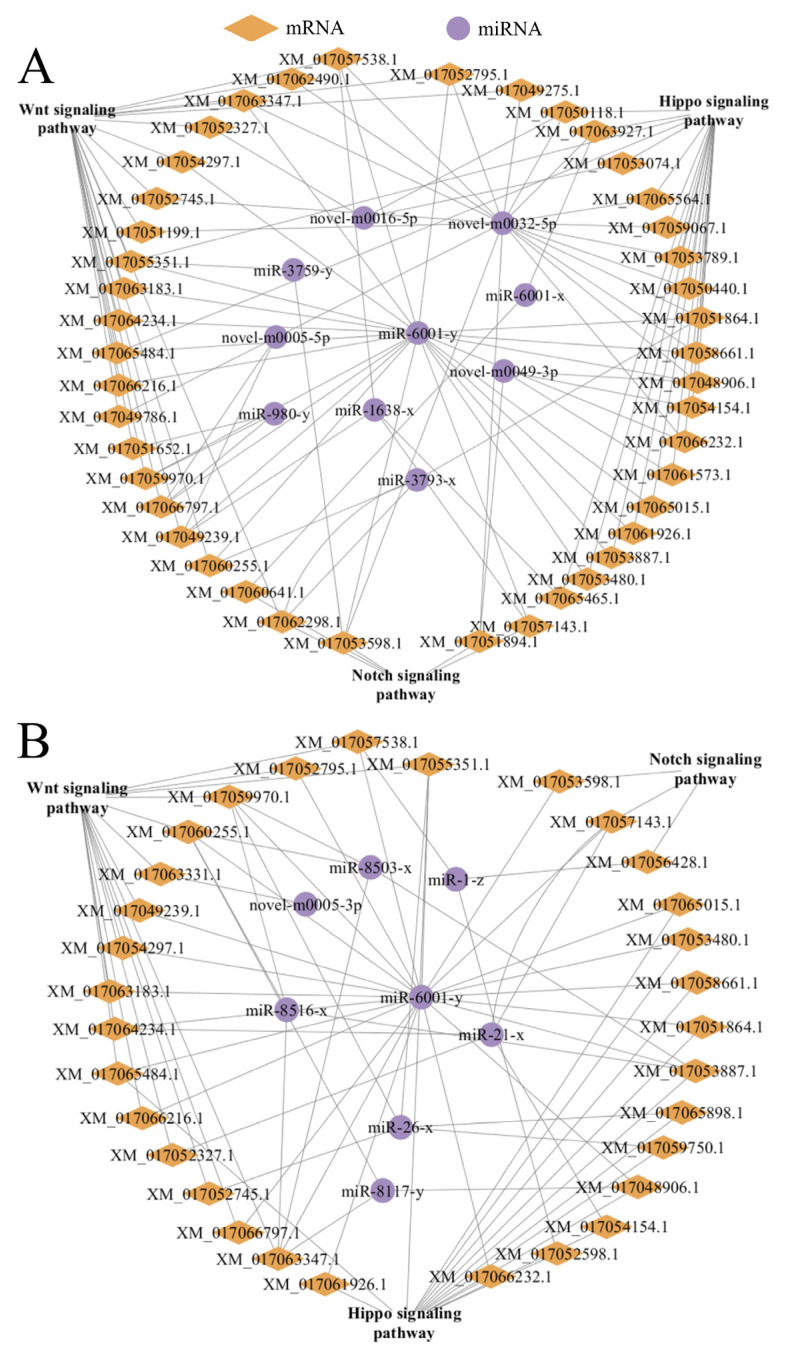
DEmiRNA–mRNA networks relevant to development-associated signaling pathways in the Ac4 vs. Ac5 (**A**) and Ac5 vs. Ac6 (**B**) comparison groups. Orange diamonds represent target mRNAs, while purple circles represent DEmiRNAs. Grey lines indicate potential targeting relationships between DEmiRNAs and target mRNAs.

**Figure 7 insects-14-00469-f007:**
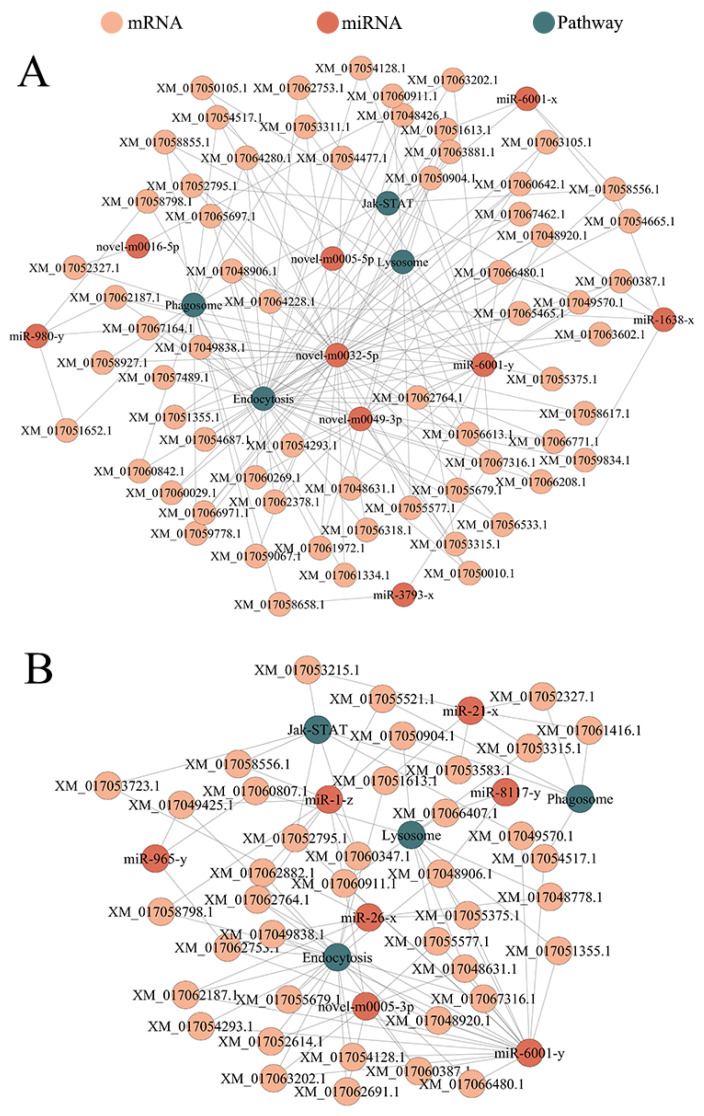
DEmiRNA-mRNA networks involved in the cellular and humoral immune in the Ac4 vs. Ac5 (**A**) and Ac4 vs. Ac5 (**B**) comparison groups. Orange circles represent target mRNAs, red circles represent miRNAs, and green circles represent annotated pathway. Grey lines indicate potential targeting relationships between the miRNAs–mRNAs pathway.

**Figure 8 insects-14-00469-f008:**
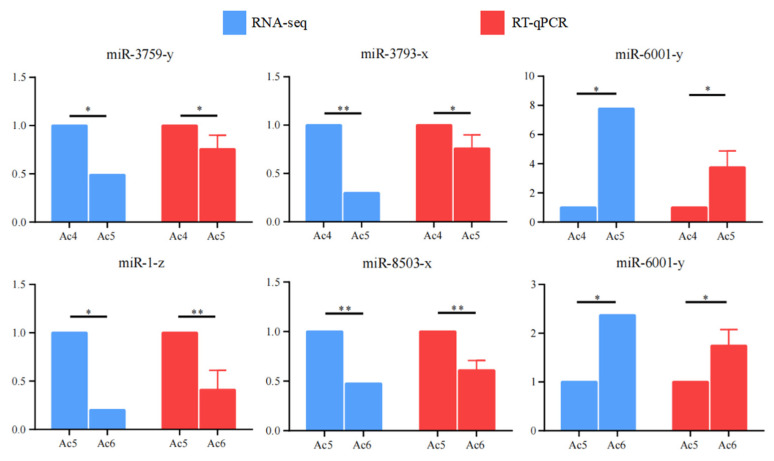
RT-qPCR validation of five DEmiRNAs. Both qPCR data and RNA-seq data were presented as mean ± standard deviation (SD) and subjected to Student’s *t* test, * *p* < 0.05; ** *p* < 0.01.

## Data Availability

All the data are contained within the article.

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
