# Peer review of "Expression Profile, Regulatory Network, and Putative Role of microRNAs in the Developmental Process of Asian Honey Bee Larval Guts"

_insects, 2023, doi:10.3390/insects14050469_

Round 1

Reviewer 1 Report

Whole manuscript is based on Bioinformatic analysis of miRNAs from their previous RNA sequence data. I would suggest authors to carry out experiments for validation of target genes against differentially expressed miRNAs as only Bioinformatics is not enough to show miRNA-mRNA interaction.

Needs rephrasing some of the sentences in introduction and discussion sections.

Author Response

1.Whole manuscript is based on Bioinformatic analysis of miRNAs from their previous RNA sequence data. I would suggest authors to carry out experiments for validation of target genes against differentially expressed miRNAs as only Bioinformatics is not enough to show miRNA-mRNA interaction.

Response: A major aim of this work was to explore the expression profile, regulatory network, and putative role of microRNAs in the developmental process of Asian honey bee larval guts, mainly based on bioinformatics. In the next step, we will perform molecular validation of DEmiRNA-mRNA interaction between several with mRNA and followed by functional investigation key DEmiRNAs and corresponding target mRNAs.

2.Needs rephrasing some of the sentences in introduction and discussion sections.

Response: Following your kind suggestion, we carefully checked the whole manuscript and tried our best to improve the written language. Thanks.

Reviewer 2 Report

Fan et al’s manuscript is a well-written entry into characterizing miRNA expression profiles: they mine a previously published RNA sequence library (d4-d6 gut from Apis cerana) for putative miRNAs, analyze differential miRNA transcript levels across d4:d5 and d5:d6, and infer putative targets and thus systems regulated by those miRNAs. While all as is described is well-designed and performed, and then clearly written, the overall contribution and value of the work is less clear. It can be expected that there would be differential transcript levels of a wide range of systems including developmentally-associated ones; similarly immune, nutrient obtainment, sensory systems, endocrine systems related to molting, etc. The authors identify these differential transcripts (Figs 1, 3) and then use bioinformatic methods to identify putative downstream modulated genes (Figs 4-7). This is somewhat like throwing a net into the water: it can be expected that some fish will be caught; here, the authors have gone fishing and unsurprisingly caught something. What is lacking is functional verification of the work: demonstration that one or more of the putative regulatory miRNAs (eg miRNA-6001-y, lines 386-89) functionally regulate the presumed downstream developmental gene would greatly enhance the work. Lacking this, the work is of moderate utility in advancing the field of understanding comparative developmental biology. This is especially true as the previous work (including library synthesis and “analysis”) was published in ref #27 Scientia Agricultura Sinica which is inaccessible to a large portion of the reading audience – what was actually analyzed previously?

Beyond this large complaint (which should not preclude publishing), there is the minor issue that the authors do not clarify (to my recollection) WHY d4, d5, d6 larval guts were analyzed. They could do this through a combination of description of physiological and developmental processes that occur through these time points (what is occurring that differs on these three days that would necessitate differential miRNA and gene expression levels?). 

There are some minor errors that should be addressed: 

Line 60-1: The authors failed to catch reference error. 

Line 70: “…with a distribution smong 18-25…” – what does this mean?

Lines 235-6: the authors fail to state the reference point for “downregulation” and “upregulation” – what is the denominator?

Lines 241: the authors fail to communicate what “cell part” refers to – while this is a GO term, it would be useful for those of us less familiar with those to know what this means

Lines 365-67: The authors state that Sgg is regulated by miRNAs, thus mediating growth. This is inferred and was not demonstrated.

Author Response

1.Fan et al’s manuscript is a well-written entry into characterizing miRNA expression profiles: they mine a previously published RNA sequence library (d4-d6 gut from Apis cerana) for putative miRNAs, analyze differential miRNA transcript levels across d4:d5 and d5:d6, and infer putative targets and thus systems regulated by those miRNAs. While all as is described is well-designed and performed, and then clearly written, the overall contribution and value of the work is less clear. It can be expected that there would be differential transcript levels of a wide range of systems including developmentally-associated ones; similarly immune, nutrient obtainment, sensory systems, endocrine systems related to molting, etc. The authors identify these differential transcripts (Figs 1, 3) and then use bioinformatic methods to identify putative downstream modulated genes (Figs 4-7). This is somewhat like throwing a net into the water: it can be expected that some fish will be caught; here, the authors have gone fishing and unsurprisingly caught something. What is lacking is functional verification of the work: demonstration that one or more of the putative regulatory miRNAs (eg miRNA-6001-y, lines 386-89) functionally regulate the presumed downstream developmental gene would greatly enhance the work. Lacking this, the work is of moderate utility in advancing the field of understanding comparative developmental biology. This is especially true as the previous work (including library synthesis and “analysis”) was published in ref #27 Scientia Agricultura Sinica which is inaccessible to a large portion of the reading audience – what was actually analyzed previously?

Response: Thank you so much for your valuable comments. In view of that knowledge of miRNAs in Asian honey bee (Apis cerana) is very limited, we previously conducted deep sequencing of the larval guts of A. cerana, (1) on basis of the clean reads after strict quality control, we first performed identification of miRNAs by mixing sequencing data from every groups in that more miRNAs could be detected based on more sequencing data, followed by confirmation of sequences and expression of some miRNAs using Stem-loop RT-PCR and Sanger sequencing, the results (published previously in Scientia Agricultura Sinica) enriched the reservoir of A. cerana miRNAs; (2) subsequently, in this work, we conducted comprehensive investigation of

the expression profile, regulatory network, and putative role of microRNAs in the developmental process of Asian honey bee larval guts; (3) in the near future, we will further conduct functional investigation of key DEmiRNAs (overexpression and knockdown via feeding method) and corresponding targets (RNAi). 

2.Beyond this large complaint (which should not preclude publishing), there is the minor issue that the authors do not clarify (to my recollection) WHY d4, d5, d6 larval guts were analyzed. They could do this through a combination of description of physiological and developmental processes that occur through these time points (what is occurring that differs on these three days that would necessitate differential miRNA and gene expression levels?).

Response: Thanks for your comment and recommendation of great importance. Here, the reasons why 4-, 5-, and 6-day-old larval guts were selected for investigation are: (1) the larval stage of a honey bee lasts 6 days old, among which 1- and 2-day-old larvae are very small and sensitive, making it difficult to obtain intact gut tissues by dissection, our preliminary experimental results showed that artificially removed 3-day-old honey bee larvae into a constant temperature and humidity chamber feeding could still maintain a high survival rate up to 6-day-old, allowing to dissect intact guts of 4-, 5-, and 6-day-old larvae; (2) in order to analyze the interaction relationship among various ncRNAs such as miRNAs, piRNAs, lncRNAs, and circRNAs, we previously conducted strand-specific cDNA library-based RNA-seq and small RNA-seq of 4-, 5-, and 6-day-old larval guts, keeping the uniform larval age in different sequencing projects. Related explanation was added in the discussion section in the revised manuscript.

3.Line 60-1: The authors failed to catch reference error.

Response: Thanks for your valuable recommendation, we carefully checked the references one by one and made necessary updates.

4.Line 70: “…with a distribution smong 18-25…” – what does this mean?

Response: “smong” here is a mistake, which should be “among”. It refers to the length distribution of the identified miRNAs.

5.Lines 235-6: the authors fail to state the reference point for “downregulation” and “upregulation” – what is the denominator?

Response: According to your helpful comment, necessary information about “downregulation” and “upregulation” was added in revised version of manuscript.

6.Lines 241: the authors fail to communicate what “cell part” refers to – while this is a GO term, it would be useful for those of us less familiar with those to know what this means

Response: Thanks for your kind suggestion, following which we added necessary explanation in the revised manuscript.

7.Lines 365-67: The authors state that Sgg is regulated by miRNAs, thus mediating growth. This is inferred and was not demonstrated.

Response: Following your helpful comment, we carefully checked and modified the corresponding descriptions in the revised version of manuscript.

Reviewer 3 Report

In this work, Fan et al. analyzed the expression profile of miRNAs during the developmental process of the larval guts of Apis cerana cerana, the nominate subspecies of Asian honey bee, Apis cerana, and further investigated the putative roles of differentially expressed miRNAs (DEmiRNAs) in the larval gut development, which offered helpful information about miRNAs in A. apis and laid a basis for illustrating the mechanism underlying DEmiRNA-regulated development of larval guts. The findings will be beneficial for researchers focusing on the developmental biology of honey bees. There were some minor points should be addressed before being considered for publication in this journal.

1. Target prediction of DEmiRNAs were performed here, but the descriptions regarding the RNA-seq data source were missing. Please provide the corresponding content.

2. The authors screened targets associated with development-related signaling pathways as well as cellular and humoral immune pathways, followed by construction of regulatory networks. The reasons for the selection should be explained in the methods section.

3. In this study, three timepoints (4-, 5-, and 6-day-old) were chosen for performing analyses. Why? This needs to be explained in the discussion section.

4. Structural characteristics of miRNAs in 4-, 5-, and 6-day-old larval guts were analyzed. Please provide corresponding discussion of this result in the discussion section.

5. P2L56: replace amount with quantity

6. P2L69: smong>among

7. P3L138: provide the reference article for the Primer Premier 6 software

8. P4L181: GenBank ID> GenBank accession number

9. P11L292: immnue>immune

10. L12P329: change An with A

Few language problems were listed in the comments to autors.

Author Response

1. Target prediction of DEmiRNAs were performed here, but the descriptions regarding the RNA-seq data source were missing. Please provide the corresponding content.

Response: According to your helpful comment, necessary information about source of DEmiRNA-targeted mRNAs was added in revised version of manuscript.

2. The authors screened targets associated with development-related signaling pathways as well as cellular and humoral immune pathways, followed by construction of regulatory networks. The reasons for the selection should be explained in the methods section.

Response: According to your helpful suggestion, we added the reasons for the selection of pathways in the revised version of manuscript.

3. In this study, three timepoints (4-, 5-, and 6-day-old) were chosen for performing analyses. Why? This needs to be explained in the discussion section.

Response: Thanks for your comments of great importance. Here, 4-, 5-, and 6-day-old larval guts were selected for investigation based on the following considerations: (1) the larval stage of honey bees lasts 6 days old, among which 1- and 2-day-old larvae are very small and sensitive, making it difficult to obtain intact gut tissues by dissection, our preliminary experimental results showed that artificially removed 3-day-old honey bee larvae into a constant temperature and humidity chamber feeding could still maintain a high survival rate up to 6-day-old, and it’s possible to prepare intact guts of 4-, 5-, and 6-day-old larvae; (2) considering that various ncRNAs such as miRNAs, piRNAs, lncRNAs, and circRNAs could mutually interact, we previously conducted RNA-seq and small RNA-seq of cDNA libraries of 4-, 5-, and 6-day-old larval guts, respectively, keeping the uniform larval age in different sequencing projects.

4. Structural characteristics of miRNAs in 4-, 5-, and 6-day-old larval guts were analyzed. Please provide corresponding discussion of this result in the discussion section.

Response: Thanks for your kind reminder and helpful comments. We added the necessary descriptions in revised manuscript.

5. P2L56: replace amount with quantity

Response: It was replaced by “quantity” in the revised manuscript.

6. P2L69: smong>among

Response: Following your kind comment, we checked and modified the related contents in the revised manuscript.

7. P3L138: provide the reference article for the Primer Premier 6 software

Response: It was modified in the revised manuscript according to your kind comment.

8. P4L181: GenBank ID> GenBank accession number

Response: Following your kind suggestion, we corrected this statement in revised manuscript.

9. P11L292: immnue>immune

Response: Correction was made following your kind comment. Thanks.

10. L12P329: change An with A

Response: This was corrected in the revised manuscript. Thanks.

Round 2

Reviewer 1 Report

I am agreed with authors response to my comments and now the manuscript is accepted for publication.